# Effects of Glycated Glutenin Heat-Processing Conditions on Its Digestibility and Induced Inflammation Levels in Cells

**DOI:** 10.3390/foods10061365

**Published:** 2021-06-12

**Authors:** Yaya Wang, Lu Dong, Yan Zhang, Junping Wang, Jin Wang, Wenwen Pang, Shuo Wang

**Affiliations:** 1Tianjin Key Laboratory of Food Science and Health, School of Medicine, Nankai University, Tianjin 300071, China; 18829349424@163.com (Y.W.); 819063@nankai.edu.cn (L.D.); yzhang@nankai.edu.cn (Y.Z.); wangjin@nankai.edu.cn (J.W.); wwpang@nankai.edu.cn (W.P.); 2College of Food Science and Engineering, Tianjin University of Science & Technology, Tianjin 300457, China; wangjp@tust.edu.cn

**Keywords:** glutenin, digestibility, heat-processing conditions, inflammation

## Abstract

Protein is one of the three major macronutrients and is essential for health. The reaction of α-dicarbonyl compounds (α-DCs) with glutenin during heat processing can modify its structure, thereby reducing its digestibility. Furthermore, advanced glycation end products (AGEs) formed by the Maillard reaction are associated with long-term diabetes-related complications. In this study, we established a heat processing reaction system for α-DCs and glutenin by simulating common food processing conditions. An in vitro digestion model was used to study the digestibility of glycated glutenin; whereupon the effects of the digestion products on macrophage inflammatory response were further investigated. It was found that reaction conditions, including temperature, treatment duration, pH, and reactant mass ratio, can significantly affect the digestibility of glycation glutenin, in which the mass ratio of reactants has the most significant influence. We demonstrated that when the mass ratio of glutenin to methylglyoxal (MGO) was 1:3, the level of inflammation induced by glycated glutenin was the highest. The mass ratio of reactants significantly affects the digestibility of glycation glutenin and the level of macrophage-induced inflammatory response. This suggests that it is possible to protect the nutritional value of protein and improve food safety by controlling the heat processing conditions of wheat products.

## 1. Introduction

Wheat is the main food in the human diet and an important source of high-quality plant protein. Glutenin, as an important protein in wheat, possesses high nutritional quality and bioavailability. Heat processing is the main method for wheat products. At high temperatures, the nutrient composition in flour will undergo complex changes. Whilst heat processing will produce certain flavor substances, it will also produce harmful substances that have effects on quality [1]. For instance, the reaction of α-dicarbonyl compounds (α-DCs) (reducing sugars and Maillard intermediates) with proteins not only results in the loss of amino acids, but also in the production of advanced glycation end products (AGEs). During food processing, these α-DCs modify the proteins in wheat, thereby altering its digestibility, thus having an effect on overall physiology.

At present, the digestibility of glycated protein in milk has been widely studied. Glycation has been shown to disrupt lysine (Lys) and arginine (Arg) residues, which are trypsin cleavage sites in intestinal digestion [2]. Wada Y et al. shows that the glycation of milk proteins during heat processing results in a decrease of its digestibility, which is mainly caused by Lys loss [3]. This is consistent with the results of Pinto et al., who shows that heating protein makes gastrointestinal tract digestion easier compared to natural β-lactoglobulin (β-Lg), while glucose causes reduced digestibility [4]. The disrupt Lys is not bioavailable, resulting in a reduction in the nutritional value of milk proteins. In addition, a study in rats showed that the real digestibility of Lys and other amino acids is reduced when fed a heated mixture of casein powder and lactose compared to unheated casein [5]. The loss of Lys in the heat processing of milk results in a significant reduction in plasma Lys levels after meals [6]. These are consistent with the results of most digestibility studies, which use an in vitro digestion model to determine the digestibility of proteins. These studies indicate that glycation during heat processing leads to a reduction in the digestibility of milk protein [4,7,8,9]. In order to further confirm these preliminary results, the digestion behavior of glycated proteins in foods other than milk should be considered. Recently, our study is the first to show that the modification of glutenin by methylglyoxal (MGO) results in a decrease in its digestibility [10]. In addition, there are few reports on the effect of glycation on the digestibility of glutenin during heat processing.

The glycation of protein in the Maillard reaction is affected by many factors, such as reaction temperature, duration, and pH—all of which alter the structure of glycated protein. Kong et al. studied the effects of different factors (molar ratio of the substrate, reaction agent, temperature, duration, and pH) on the degree of glycation and the structural proteins [11]. The results show that the above factors all have significant effects on the degree of glycation and the structure of glycated β-Lg. Of all the factors, MGO results in the greatest degree of glycation of glycated β-Lg. With the increase of temperature (55–95 °C) and duration (0–120 min), the degree of glycation of β-Lg increases. Current studies have shown that glycation of proteins in the heating process will eventually produce AGEs, and that reaction conditions have an important effect on the degree of glycation and the type of products. However, there are few relevant studies on how the reaction conditions affect the digestibility of glycated proteins, especially for glutenin, which has not been reported.

The physiological consequences of dietary AGEs are not fully understood. However, endogenous production of AGEs in diabetics has been shown to stimulate inflammation and contribute to the onset of cardiovascular diseases, such as atherosclerosis and diabetic cardiomyopathy [12]. Inflammation in these conditions is thought to be induced by AGEs binding to AGE-specific receptors [13]. Although inflammation caused by endogenous AGEs has been well studied, there is little research on inflammation induced by dietary AGEs. Recently, Timme et al. found that dietary AGEs can cause human macrophage inflammatory response in vitro by binding to RAGE receptors [14]. However, different molecular weights of AGE show significant differences in their interaction with AGE receptors [15]. Dietary AGEs are hydrolyzed into smaller parts by digestive enzymes and then actively transported into the body’s circulation [16,17]. Therefore, we hypothesized that processing conditions would affect the digestibility of glycation glutenin and the molecular weight of the digested products, thus affecting its binding to RAGE receptor and the degree of inflammatory response.

Therefore, the purpose of this study is to investigate the effects of different processing conditions on the digestibility of glycated glutenin and the inflammatory response caused by its digestible products. The results of this study are expected to provide evidence for reducing nutrient loss of glutenin during heating, improving digestibility, and reducing inflammation risk factors by controlling heating.

## 2. Materials and Methods

### 2.1. Chemicals and Materials

The wheat used was purchased from a local commercial market. All water used in the experiments was produced from a Milli-Q Ultrapure Water Systems, and all chemicals used were analytical grade, unless specified. Pepsin from porcine gastric mucosa (>2500 U/mg), trypsin from porcine pancreas (1655 U/mg), and chymotrypsin (>40 U/mg) were purchased from the Sigma-Aldrich Chemical Corporation. MGO and glyoxal (GO) were purchased from Sigma-Aldrich Chemical Company (St. Louis, MO, USA).

### 2.2. Separation of Glutenin

Glutenin was extracted from wheat flour, as previously described [18]. Briefly, the gluten was obtained by washing the dough with ultra-pure water. n-hexane was added to the gluten in a ratio of 1:20 (n-hexane: gluten; *w*/*v*). The gluten was stirred at room temperature for 1 h and then suspended in the fumigation cabinet overnight. A total of 0.4 mol/L NaCl solution was added to the gluten in a ratio of 1:20 (NaCl: gluten; *w*/*v*) and stirred again for 1 h at room temperature. Then, the suspension was centrifuged to collect the precipitation. The ultrapure water was added to the precipitate in a ratio of 1:20 (*w*/*v*) and stirred at room temperature for 1 h. The precipitate was centrifuged and continued to be collected. The precipitate was dissolved again in 70% alcohol and stirred at room temperature for 1 h. The precipitate was centrifuged and collected. Each extraction step was repeated three times. The final precipitate was freeze-dried at −80 °C and grounded into a powder to obtain natural glutenin.

### 2.3. Preparation of Glutenin Glycation System for Influencing Factor Analysis

The glutenin was incubated with MGO and GO separately to determine several factors. Firstly, the effect of reactant mass ratio on the digestibility of glutenin was determined. Glutenin (1 mg/mL) with MGO (40% aqueous solution) and GO was incubated at different mass ratio of glutenin to MGO or GO (1:0, 1:0.5, 1:1, 1:2, 1:5) at 100 °C for 8, 16, 24, 32, 40, and 48 min, respectively. Secondly, the effect of temperature on the digestibility of glutenin was detected. Glutenin (1 mg/mL) with MGO (40% aqueous solution) and GO was incubated at different temperatures (80, 100, 120, 140 °C) for 8, 16, 24, 32, 40, and 48 min, respectively. In the above studies, the mass ratio of glutenin to MGO or GO was 1:5. Next, the effect of pH on the digestibility of glutenin was studied. Glutenin (1 mg/mL) with MGO (40% aqueous solution) and GO was incubated in different pH solutions (6.0, 6.5, 7.0, 7.5, 8.0) at 100 °C for 8, 16, 24, 32, 40, and 48 min, respectively. Similarly, the mass ratio of gluten to MGO or GO is 1:5. After the heat treatment, the samples, from each different time point, were cooled down in an ice bath (about 0 °C) to stop the reaction.

### 2.4. Simulated Digestion of the Sample

The digestibility of glutenin was measured according to the method described by Picariello et al. [19] with slight modifications. A sample of 5 mg glutenin was accurately weighed and placed in a centrifuge tube. A total of 3.5 mL of simulated gastric juice (35 mM NaCl, 200 U/mg pepsin, pH 2.0) was added to each tube, and the mixture was incubated at 37 °C for 1 h. After incubation, 145 μL 1 M NaHCO_3_ was added to adjust pH to neutral to terminate the reaction. Then, 1 mL of simulated intestinal digestive solution (406 μL Bis-Tris, 38 μL CaCl_2_, 148 μL bile salt, 40 U/mg trypsin, and 0.5 U/mg chymotrypsin) was added, mixed, and incubated at 37 °C for 2 h. After incubation, the reaction was terminated by heating in boiling water at 100 ℃ for 5 min. The digested tubes were cooled to room temperature and centrifuged (9000× *g*, 4 °C, 5 min), the supernatants were collected to quantify the DH (degree of hydrolysis).

### 2.5. Determination of Digestibility

The DH was determined according to Wu et al [20]. Tryptophan was used as the standard in this experiment. A standard curve was established with a series of different concentrations of tryptophan (0.001, 0.005, 0.010, 0.020, 0.050, 0.060 mmol/mL). The samples of digestive products in this experiment need to be diluted 3 times. A total of 400 uL of the diluted sample solution was added to a centrifuge tube containing 3 mL OPA (100 mM sodium tetraborate, 0.01% SDS, 0.05 mg/mL o-phthalaldehyde, 0.05 mg/mL dithiothreitol), and the solution was mixed and reacted for 2 min without light. The fluorescence emission (excitation: 340 nm, emission: 450 nm) was measured by a microplate analyzer (Varioskan LUX, Thermo Scientific, Waltham, MA, USA). Three parallel experiments were carried out for each sample. At the same time, blank control tests were carried out, and unhydrolyzed samples were substituted. Finally, the degree of hydrolysis was calculated according to the standard curve. The DH was calculated using the following equation:$$DH\%=\frac{h_{s}}{h_{total}}\times 100\%$$
*h_s_* is defined as the mmol of free amino groups per gram of protein in the sample, and *h_total_* is the mmol of free amino groups per gram of protein, assuming complete hydrolysis of the protein (7.96 mmol/g protein).

### 2.6. Matrix-Assisted Laser Desorption/Ionization Time of Flight Mass Spectrometry (MAL-DI-TOF-MS) Analysis of Sample Digest

The ZipTip C18 pipette tip [21] was pre-equilibrated with acetonitrile and 0.1% trifluoroacetic acid solution. Then, the peptides in the digestion solution were absorbed on the C18 column by repeated blowing with 10 μL of digestive solution, and then washed with 0.1% trifluoroacetic acid solution for 2–3 times to remove the salt. The target peptide was then eluted with 20% acetonitrile solution. A total of 1 μL desalted sample was added to the ground steel BC target (MTP 384, Bruker, Bremen, Germany) and allowed to dry naturally. A matrix solution of 1 uL CHCA (α-Cyano-4-Hydroxycinnamic Acid) was then added to the sample for crystallization, and the target plate was placed in the instrument for detection after the matrix dried. MALDI-TOF mass analysis was performed on the Ultrafle XTreme TOF-TOF mass spectrometer (Brook, Bremen, Germany). The detection mode was reflective positive ion mode, the *m/z* detection range was 0–5000, and the laser intensity was 60%. The experimental data was analyzed by Flex Analysis Batch Process software (13.0 Bruker, Bremen, Germany).

### 2.7. Cell Culture

Mouse RAW264.7 macrophages (purchased from American Type Collection) were cultured in RPMI 1640 supplemented with 10% FBS plus 1% PS (Gibco, Thermo Scientific, Waltham, MA, USA). The cells were placed in a cell incubator containing 5% carbon dioxide for culture passage and experiment. The cells used in this study were from the fourth generation.

### 2.8. Total RNA Isolation and RT-qPCR

RAW264.7 cells (106 cells/well) were placed in a 6-well plate and cultured for 24 h. After aspirating the medium, fresh medium supplemented with glutenin or glycated glutenin digestion products (10%, *v*/*v*) was added to the 6-well plate and cultured for 24 h. After washing the wells with PBS, the total RNA was extracted with the TRIzo1 (Thermo Fisher Sientific, 81 Wyman Street, Waltham, US) method. The microplate reader was used to determine the purity of the extracted RNA to ensure that the absorbance of the sample was between 1.8–2.1. LunaScriptTM SuperMix Kit (New England BioLabs, Ipswich, MA, USA) was used for cDNA synthesis. Finally, the genes that encode β-actin were used as housekeeping genes. According to Yu’s report [22], the primer sequences of β-actin, IL-6, TNF-α, IL-1β, and RAGE were designed. The primers are shown in Appendix A.

### 2.9. Statistical Analysis

SPSS 21.0 was employed to perform the statistical analysis, and all the results are expressed herein as the mean ± standard deviation. Multi-way ANOVA with a general linear model was selected to investigate factor effects on the digestibility of glycated glutenin and potential interactions among them. Significant differences among treatments were compared using Tukey’s test. All results were considered statistically significant at *p* < 0.05.

## 3. Results

### 3.1. Effect of Reactant Mass Ratio and Duration on the Digestibility of Glycated Glutenin Derived from MGO and GO

The effect of reactant mass ratio and duration on the digestibility of glycated gluten are shown in Figure 1. The mass ratio of reactants has a significant impact on digestibility. The smaller the mass ratio of glutenin to MGO/GO, the lower the digestibility is, regardless of the duration. However, in the Glutenin-MGO/GO system, when the heating temperature is constant, the duration has no obvious effect on the digestibility of glycated glutenin. In addition, under the same heat processing conditions, the digestibility of glycated glutenin derived from MGO is less than glycated glutenin derived from GO.

### 3.2. Effect of Temperature and Duration on the Digestibility of Glycated Glutenin Derived from MGO and GO

The effect of temperature and duration on the digestibility of glycated glutenin are shown in Figure 2. The digestibility of glycated glutenin derived from GO decreases with an increase in temperature, regardless of duration. However, the digestibility of glycated glutenin derived from MGO is determined by the combination of duration and temperature. When the duration is 8–16 min, the digestibility of glycated glutenin derived from MGO decreases with increasing temperature, and its digestibility increases with increasing temperature during the duration of 24–48 min. It was also observed that the digestibility of glycated glutenin derived from GO decreased with duration and the digestibility of glycated glutenin derived from MGO tends to increase with duration at the same temperature.

### 3.3. Effect of pH and Duration on the Digestibility of Glycated Glutenin Derived from MGO and GO

The effect of pH and duration on the digestibility of glycated glutenin derived from MGO and GO are shown in Figure 3. The digestibility of glycated glutenin derived from GO decreases as the pH buffer changes from acidic to alkaline, while the digestibility of glycated glutenin derived from MGO increases first and then decreases. Under acidic conditions, the digestibility of glycated glutenin derived from GO gradually decreases with the increase of duration, but its digestibility does not change significantly under neutral and alkaline conditions. However, under the same pH value, the glycated glutenin derived from MGO does not change significantly with the duration.

### 3.4. Relative Contributions of Factors and Interactions to the Effect of Glycated Glutenin on Digestibility

The percentage contribution of each factor and associated interaction is calculated using the method described by Chen and Kitts, et al. [23], to determine the relative importance of the reaction factors, such as temperature, pH, and reactant mass ratio, on the digestibility of glycated glutenin. (Table 1) The mass ratio of reactants has the greatest impact on the digestibility of glycated glutenin. In the glutenin-MGO/GO system, the digestibility of glycated glutenin is most affected by the mass ratio of reactants, followed by temperature, then pH.

### 3.5. Effect of Reactant Mass Ratio on Inflammation in RAW264.7 Cells and Expression of RAGE Receptor Induced by Glycated Glutenin Derived from MGO

A large number of in vivo and in vitro studies have shown that the stimulation of exogenous substances can promote the release of excessive pro-inflammatory cytokines from macrophages. Therefore, in order to explore the effect of reactant ratio on the level of inflammation in macrophages induced by glycated glutenin, qPCR is used to determine the mRNA expression of the inflammation markers IL-6, TNF-, and IL-1β. As shown in Figure 4a, when the mass ratio of glutenin to MGO is 1:2, the expression of IL-6 is relatively high; however, when the mass ratio of glutenin to MGO is 1:3, the expression of TNF-a and IL-1β is higher, as shown in Figure 4b,c. The above results indicate that when the ratio of glutenin to MGO is at an intermediate value, the level of cellular inflammation induced by glycated glutenin is higher, which implies that the mass ratio of reactants affects the expression of cellular inflammation marker mRNA. AGEs-dependent receptor changes are believed to play an important role in the development of many chronic inflammations. Therefore, we analyzed the mRNA expression of RAGE to study the mechanism of inflammation marker changes. As shown in Figure 4d, when the mass ratio of glutenin to MGO is 1:3, the mRNA expression of RAGE is the highest. According to the data of mRNA expression and inflammation markers, it can be inferred that different reactant mass ratios may affect the state of cell inflammation by affecting the activation of RAGE.

### 3.6. Peptide Map Analysis of Glycated Glutenin Derived from MGO in Different Reactants Mass Ratios

MALDI-TOF was used to determine the molecular weight of the digested products of glycated glutenin at different mass ratios of MGO to glutenin. As shown in Figure 5, the molecular weight of the digested product of natural glutenin is about 700–2400 Da. The molecular weight of glycated glutenin derived from MGO digestion products decreases significantly with the increase in the mass ratio of MGO to glutenin. When the mass ratio of MGO to glutenin is 1:5, the molecular weight distribution of glycated glutenin derived from MGO digestion products is between 800–950 Da.

## 4. Discussion

Over the past few decades, the consumption of heat processed foods has increased exponentially. Heat processing is the most widely used food processing technology with the longest history, constantly promoting the development of the food industry, and providing consumers with an abundant food supply. Heat processing technology is used to improve food flavor and texture, extend shelf life, and ensure food safety, and Maillard reactions often accompany this process. In wheat products, gluten, starch, and sugar are the main participants of the Maillard reaction, and the complex reactions among them have always been the focus of research in the field of food processing. AGEs have attracted a lot of attention because of the health risks associated with their reaction products. α-DCs are very important active intermediates in the Maillard reaction of sugars during heat processing, which can modify proteins to form AGEs. The formation of AGEs is usually accompanied by protein structural changes, which are closely related to the digestibility and bioavailability of proteins. The glycation of milk protein as a result of heat treatment, especially spray drying and reheating, is believed to reduce its nutritional quality [24]. The different states of proteins after heat treatment may have a great influence on protein digestion. This is regulated by structural and chemical modifications of the protein, both of which may affect the accessibility of amino acid bonds to proteolytic enzymes. Sheng et al. studied the effects of glycation of GO, MGO, and diacetyl on the digestibility of BSA in vitro, and the results show that glycation of α-DCs would reduce the digestibility of BSA in stomach and intestinal digestion [25]. In addition, several recent reports have investigated the effects of α-DCs on protein digestibility in milk. It was found that glycation of α-DCs reduced in vitro digestibility of β-casein and β-Lg in milk at the gastric and intestinal stages [26]. The effects of glycation of milk proteins on their digestibility have been extensively studied by inducing glycation using different products at different heating modes and temperatures. It was found that the effect of glycation on the digestibility of milk protein is closely related to the conditions of heat processing [25,26,27]. Glutenin is one of the most important proteins in wheat, and it is of great significance to study the effect of glycation on its digestibility during heat processing. Our previous work reported the effect of different processing temperatures on the digestibility of glycated glutenin caused by α-DCs. Digestibility of glycated protein depends on a number of factors, including temperature, duration, pH, and the glycosylating agent. The results showed that α-DCs could reduce the digestibility of glutenin, which decreased with increasing temperature [10]. In order to further confirm these preliminary results, the influence of α-DCs on the digestibility of glutenin under other heat processing conditions should be considered.

In this study, we explored the influence of mass ratio of reactants, reactant type, temperature, duration, and pH on the digestibility of glycated glutenin derived from α-DCs. MGO and GO are the two most common α-DCs in food. We separately explored their influence on the digestibility of glycated glutenin under different mass ratios of reactants, temperature, and pH. In addition, it is generally believed that the duration has a profound effect on the degree of Maillard reaction. However, the processing duration of different flour products are different; for example, pasta takes about 8–10 min to cook, while cakes take about 30–40 min to bake. In order to comprehensively explore the effects of different treatment durations on the digestibility of glycated glutenin, we separately explored the effects of reactant mass ratio, temperature, pH, and duration on the digestibility of glycated glutenin. Among them, the mass ratio of reactants has the greatest impact on the digestibility of glycated glutenin. This result suggests that the content of α-DCs in food has a significant impact on protein digestibility. As shown in Figure 1, the digestibility of glutenin decreases as the mass ratio of MGO to glutenin increases. The decrease in digestibility is due to the glycation derived from α-DCs, which disrupt Lys and Arg residues, thus reducing their sensitivity to proteases in the gastrointestinal tract. Therefore, the content of α-DCs in food systems directly determines the degree of disruption of Lys and Arg residues, which determines the digestibility of protein [2].

Temperature is also an important factor affecting the Maillard reaction [2]. According to experience, within a certain period of duration, if the temperature of the reaction system increases by 10 °C, the Maillard reaction speed will double. Broersen et al. reported that an increase of 15 °C in temperature results in a 56% increase in the disrupted Lys and Arg content in the conjugate formed between β-Lg and glucose [28]. Likewise, an increase of 10 °C resulted in a 62% increase in Lys and Arg disrupted in the conjugate produced between β-Lg and tagatose [29]. Our research results show that the digestibility of glycated glutenin derived from GO decreases with the increase of temperature during a treatment duration of 8–48 min. However, when the duration is 8–16 min, the digestibility of glycated glutenin derived from MGO decreases with increasing temperature. Though when treated for 24–48 min, its digestibility increases with increasing temperature. Although our research specifically focuses on glycation, all studies are conducted under heating conditions. To some degree, heating proteins improves their digestibility because of heat-derived protein denaturation and better protease accessibility [24]. However, simultaneous glycation can block certain amino acids and reduce their bioavailability [2]. The blocked amino acids may greatly reduce the sensitivity of the glycated protein to digestive proteases, thereby impairing protein digestibility. In addition, a new structure can be formed and cross-linked. The cross-linked glycated structure has reduced flexibility, thus impeding the action of digestive enzymes by blocking cleavage sites and steric hindrances [17,27]. Wada and Lonnerdal digested heated dairy products in vitro and in rats, respectively, and found that pasteurization and in-can sterilization led to denaturation and aggregation [3] Carbonaro and Lindberg et al.‘s studies have similar results [30,31]. They suggest that heat-induced denaturation improves protein digestibility, while aggregation reduces it. This is consistent with the results of Pinto et al., which showed that natural β-Lg is difficult to digest, whereas heated β-Lg is readily digested, and the addition of glucose limits this heat-induced increase in digestibility [4]. This suggests that a balance of thermal denaturation and glycation affects the digestion of β-Lg. When the duration is short and the temperature increases, the modification of glutenin by MGO may lead to the production of a dimer, and these products are still indigestible with pepsin in most cases [32,33]. With prolonged heating, other aggregates form as the temperature rises, including unnatural aggregates connected by disulfide bonds, which are quickly digested by pepsin. This is consistent with the results of Peram et al.‘s study on milk [34]. In general, our findings indicate that the interaction between temperature and duration determines the digestibility of glycated glutenin to the greatest extent.

It is generally believed that pH can profoundly affect the reaction rate and pathway of Maillard reaction, thus affecting AGEs production and protein structure changes [35]. The initial pH of the reactants and the buffering capacity of the system affect the rate of the Maillard reaction [36]. Different types of pasta products differ in pH, and the same type of pasta products will also undergo great pH changes during processing. For example, the pH of the dough changes between 5–7 during the fermentation process. The rate of the Maillard reaction is considered to be very low at acidic pH, but it increases as the pH increases. Our study shows that with the increase of pH, the digestibility of glycated glutenin derived from GO decreases, while the digestibility of glycated glutenin derived from MGO first increases and then decreases. Evidently, pH values have different effects on the digestibility of glycated glutenin derived from different α-DCs. The reactivity of MGO is higher than that of GO. It is speculated that in the reaction system of GO and glutenin, the reaction of GO modifying the amino side chain of glutenin is dominant. The glycation reaction increases with the increase of pH, which leads to more restriction sites being destroyed and masked, resulting in reduced digestibility. However, in the reaction system between MGO and glutenin, the heat-induced thermal deformation is dominant after MGO is consumed by the amino side chain of glutenin. Thus, the digestibility increases as the pH increases. In addition, the digestibility of glycated glutenin derived from GO decreases with treatment duration under acidic conditions, but it does not change significantly under neutral and alkaline conditions, indicating that the interaction between pH value and duration determines the digestibility of glycated glutenin.

We found that the glycation modification of glutenin by α-DCs can change its digestibility, and this change is closely related to heat processing conditions. The change of glutenin digestibility is ultimately caused by its structural changes, which may include cross-linking, aggregation, and destruction of restriction sites. Therefore, under different heat processing conditions, α-DCs-modified glutenin may form different glycated glutenin structures.

An increasing number of research results show that the intake of foods with high AGEs is related to oxidative stress, inflammation, and cardiovascular risks in both healthy individuals and patients with chronic diseases [36,37,38]. Consequentially, dietary AGEs have become a hot issue in the food industry. Studies have found that different structures of dietary AGEs have different behaviors in gastrointestinal digestion, intestinal absorption, interaction with AGEs receptors, internal circulation, detoxification, and renal clearance [15]. Recently, mice fed with heat-processed diets with high bound AGEs show higher amounts of proinflammatory cytokines including IL-16, IL-1α, ICAM, TIMP-1, and C5a. By contrast, the unprocessed diet with a high content of free CML does not induce inflammation of comparable severity, implying that free AGEs are not efficient to activate RAGE-mediated signaling cascades [38]. It implies that whether dietary AGEs can effectively activate RAGE and induce inflammation is closely related to its molecular size and structure. In addition, under different conditions, the peptide spectra of glycated β-casein and β-lactoglobulin digestion products by α-DCs are significantly different [27], indicating that peptide release induced by glycated protein is different, and that processing conditions affect the structure of digested peptides of glycated protein. The effect of heat processing conditions on the inflammatory response induced by digested products of glycated glutenin has not been studied yet. Given that the above research results show that the mass ratio of reactants has the greatest impact on glycated protein digestibility, we further studied the influence of MGO and glutenin mass ratio on the inflammation marker levels induced by glycated protein digestion products. Our results show that when the mass ratio of glutenin to MGO is in the intermediate range, the level of cellular inflammation induced by glycation of glutenin is the highest. The mRNA expression of RAGE is also highest when the mass ratio of glutenin to MGO is in this range. Therefore, it can be inferred that different reactant mass ratios may influence the inflammatory state of cells by affecting the activation of RAGE. The results of this study show that the molecular weight of the digested product of glycated glutenin derived from MGO increases with the increase in the mass ratio of MGO to glutenin. This is possibly due to the fact that only AGEs with a specific structure and molecular weight can bind to RAGE. Molecular weights too high or too low are not conducive to the binding of AGEs to RAGE.

## 5. Conclusions

In this study, for the first time, the effect of different heat processing conditions on the digestibility of glycated glutenin was explored. The present study has found that the digestibility of glycated glutenin was greatly impacted by the temperature, treatment duration, pH, and mass ratio of the reactants, wherein the mass ratio of the reactants has the greatest impact. Moreover, different combinations of temperature and treatment duration significantly influence the digestibility of glycated glutenin. Further research is needed to explore why the digestibility of glycated glutenin is more affected by the reaction temperature and duration in glutenin-MGO than the pH value. It is difficult to estimate, however, whether and to what extent temperature is important in complex food processing. This is only a preliminary study into the glycation of glutenin in various conditions. Moreover, we demonstrated that the reactant mass ratio also had an influence on the degree of AGE-induced macrophage inflammation in cell experiments. The results of this study provide a basis for reducing nutrient loss of glutenin during heating, improving digestibility, and reducing inflammation risk factors by controlling heating. In further studies we will verify the influence mechanism of different heat processing conditions on the digestibility of glycated glutenin.

## Figures and Tables

**Figure 1 foods-10-01365-f001:**
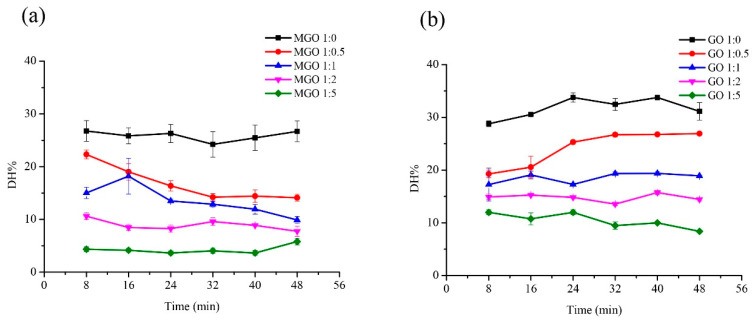
Effect of reactant mass ratio and duration on the digestibility of glycated glutenin derived from MGO and GO. (**a**) Glutenin (1 mg/mL) with MGO (40% aqueous solution) was incubated under different mass ratio of glutenin to MGO (1:0, 1:0.5, 1:1, 1:2, 1:5) at 100 °C for 8, 16, 24, 32, 40, and 48 min, respectively. (**b**) Glutenin (1 mg/mL) with GO was incubated under different mass ratio of glutenin to GO (1:0, 1:0.5, 1:1, 1:2, 1:5) at 100 °C for 8, 16, 24, 32, 40, and 48 min, respectively. Data show the means ± SD of triplicate independent experiments. Bars represent the standard error of the mean.

**Figure 2 foods-10-01365-f002:**
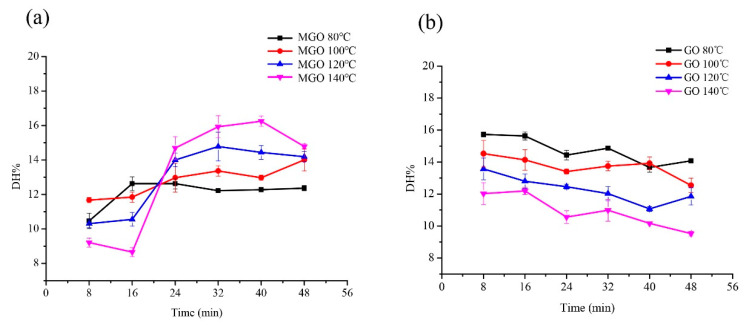
Effect of temperature and duration on the digestibility of glycated glutenin derived from MGO and GO. (**a**) Glutenin (1 mg/mL) with MGO (40% aqueous solution) was incubated for 8, 16, 24, 32, 40, and 48 min at different temperatures (80, 100, 120, and 140 °C) with the mass ratio of glutenin to MGO of 1:5. (**b**) Glutenin (1 mg/mL) with GO was incubated for 8, 16, 24, 32, 40, and 48 min at different temperatures (80, 100, 120, and 140 °C) with the mass ratio of glutenin to GO of 1:5. Data show the means ± SD of triplicate independent experiments. Bars represent the standard error of the mean.

**Figure 3 foods-10-01365-f003:**
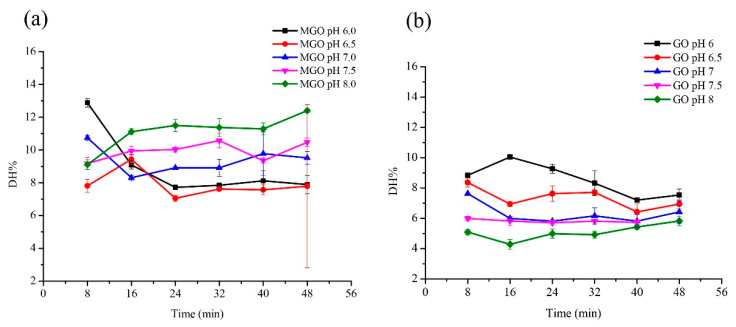
Effect of pH and duration on the digestibility of glycated glutenin derived from MGO and GO. (**a**) Glutenin (1 mg/mL) with MGO (40% aqueous solution) was incubated in different pH solutions (6.0, 6.5, 7.0, 7.5, 8.0) at 100 °C for 8, 16, 24, 32, 40, and 48 min with the mass ratio of glutenin to GO of 1:5. (**b**) Glutenin (1 mg/mL) with GO was incubated in different pH solutions (6.0, 6.5, 7.0, 7.5, 8.0) at 100 °C for 8, 16, 24, 32, 40, and 48 min with the mass ratio of glutenin to GO of 1:5. Data show the means ± SD of triplicate independent experiments. Bars represent the standard error of the mean.

**Figure 4 foods-10-01365-f004:**
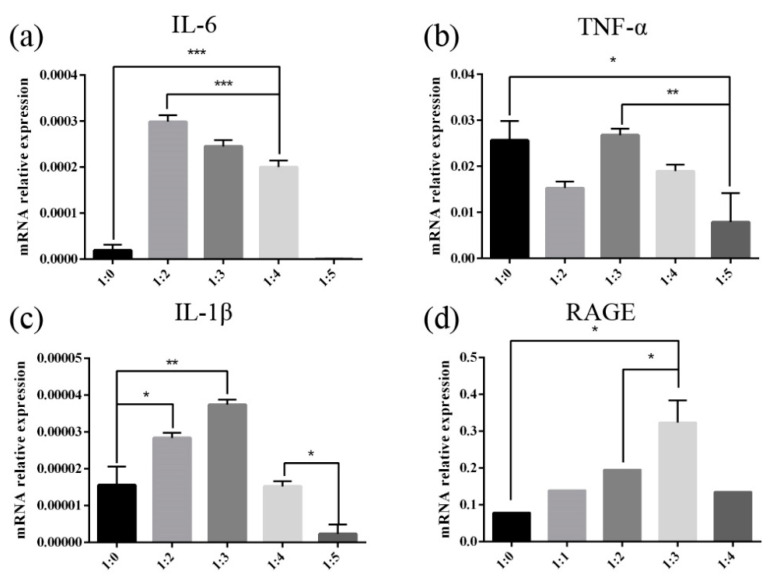
The mRNA expression levels of (**a**) IL-6, (**b**) TNF-α, (**c**) IL-1β, (**d**) RAGE. The expression levels of IL-6, TNF-α, IL-1β, and RAGE are detected in the cell supernatant. Data are shown as the mean ± SD. Data are presented as mean ± SEM. Significant differences of IL-6, TNF-α, IL-1β, RAGE are indicated as * *p* < 0.05, ** *p* < 0.01, *** *p* < 0.001.

**Figure 5 foods-10-01365-f005:**
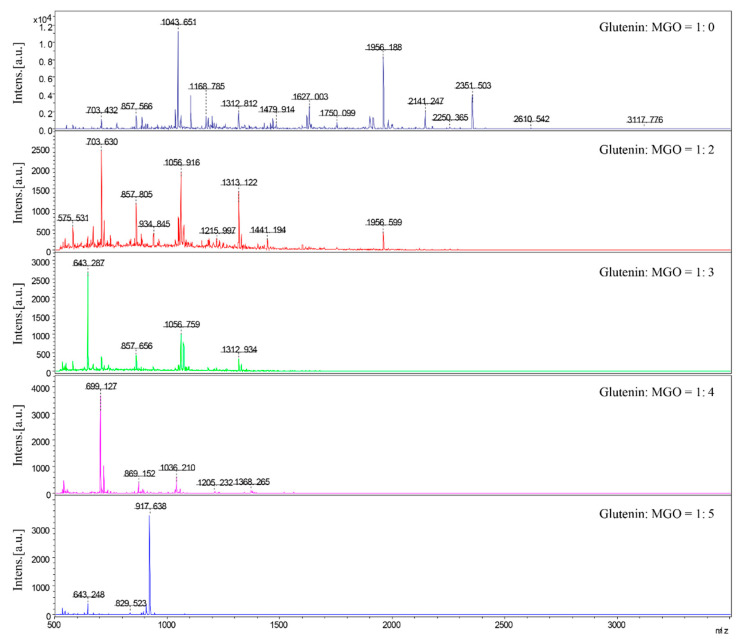
Peptide map analysis of glycated glutenin derived from MGO in different reactants mass ratios.

**Table 1 foods-10-01365-t001:** Relative contributions^1^ of factors and interactions to the effect of glycated glutenin on digestibility.

Source	SS		df		MS		F		P		Contribution (%)	
	MGO	GO	MGO	GO	MGO	GO	MGO	GO	MGO	GO	MGO	GO
Treatment duration	6.573	4.263	5	5	1.315	0.853	0.303	0.060	0.902	0.997	2.58	0.14
Reactant mass ratio	1.956	2507.197	4	4	0.489	626.799	0.113	43.783	0.976	0.000	0.77	82.97
pH	54.988	144.001	4	4	13.747	36.000	3.167	2.515	0.054	0.097	21.60	4.77
Temperature	38.122	101.143	3	3	12.707	33.714	2.928	2.355	0.077	0.123	14.97	3.35
Treatment duration * reactant mass ratio	11.971	82.720	20	20	0.599	4.136	0.138	0.289	1.000	0.993	4.70	2.74
Treatment duration * pH	32.105	7.115	20	20	1.605	0.356	0.370	0.025	0.976	1.000	12.61	0.24
Treatment duration * temperature	56.819	3.462	15	15	3.788	0.231	0.873	0.016	0.604	1.000	22.32	0.11
Error	52.083	171.794	12	12	4.340	14.316	-	-	-	-	20.46	5.69

^1^ Contribution (%) = sum of squared deviations of each factor/total sum of squared deviations. * 100.

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
