# Peer review of "Effects of Glycated Glutenin Heat-Processing Conditions on Its Digestibility and Induced Inflammation Levels in Cells"

_foods, 2021, doi:10.3390/foods10061365_

Round 1

Reviewer 1 Report

The manuscript titled “Effects of Heat-processing Conditions on the Digestibility and the Level of Induced Inflammation Levels in the cells of Glycated Glutenin” deals within the scope of the Foods Journal, by investigating an interesting topic of research. The quality of the research work presented is very good and represents a valuable result of the glycated glutenin digestion analysis.

Therefore, I would suggest minor corrections.

Please find below some remarks to help the revision of the manuscript.

Line 1: “Type of paper” should be Article.

Lines 13, 14, 21: Abbreviations should be defined the first time they appear in each of three sections: the abstract; the main text; the first figure or table.

Line 96: Lines121-125: “Glutenin was extracted from wheat flour, as previously described [1].” The authors refer to the research of Osborne et al. Therefore, this paper (book) should be cited.

Line 109: The authors should explain what “… different reactants (MGO and GO)” mean and how they were obtained. They are not mentioned in the Chemicals and Materials section.

Lines 114 and 116: Ratio 1:1?

Line 150: Please, define the abbreviation.

Line 197 (Figure 1): Figures a) and b) appear to have been replaced.

Line 216 (Figure 2): Same as previous.

Line 232 (Figure 3): Same as previous.

Line 239 (Table 1): The authors should provide the complete table of ANOVA results with degrees of freedom, sum of squares, mean squares, F and p values.

Line 445: “6. Patents”???

Author Response

Dear Reviewer:

Thank you very much for your letter and comments on our manuscript entitled “Effects of Heat-processing Conditions on the Digestibility and the Level of Induced Inflammation Levels in the cells of Glycated Glutenin” (ID: foods-1220025). We greatly appreciate the constructive comments that are very helpful for our revision of the manuscript. We have made additions and corrections according to your comments. The revised sections in the manuscript are highlighted in red. We have commissioned native English speakers to revise the language of the manuscript to make it more concise and accurate.

We do hope that the revised manuscript adequately addressed your comments. The major corrections and the response to the reviewers’ comments are as follows:

Comment 1: Line 1: “Type of paper” should be Article.

Reply to Comment 1: Thanks! We are very sorry for our negligence, and we have specified the type of paper in revised manuscript. (See Revised Manuscript, Page 1. Line 1)

Comment 2: Lines 13, 14, 21: Abbreviations should be defined the first time they appear in each of three sections: the abstract; the main text; the first figure or table.

Reply to Comment 2: Thank you very much for your kind reminder. We have specified α-DCs, AGEs and MGO in revised manuscript. (See Revised Manuscript, Page 1. Line 12, 13, 21)

Comment 3: Line 96: Lines121-125: “Glutenin was extracted from wheat flour, as previously described [1].” The authors refer to the research of Osborne et al. Therefore, this paper (book) should be cited.

Reply to Comment 3: Thank you very much for your kind reminder. The research of Osborne et al. has been cited. (See Revised Manuscript, Page 3. Line 99)

Comment 4: Line 109: The authors should explain what “… different reactants (MGO and GO)” mean and how they were obtained. They are not mentioned in the Chemicals and Materials section.

Reply to Comment 4: Thank you very much for your kind reminder. We are very sorry for our negligence, and we have supplemented and revised the manuscript. (See Revised Manuscript, Page 3. Line 96, 112)

Comment 5: Lines 114 and 116: Ratio 1:1?

Reply to Comment 5: Thank you very much! We are very sorry for our negligence. We have added the ratio of MGO and glutenin in revised manuscript. (See Revised Manuscript, Page 3. Line 119)

Comment 6: Line 150: Please, define the abbreviation.

Reply to Comment 6: Special thanks to you for your comments! “MALDI-TOF-MS” is the abbreviation of Matrix-Assisted Laser Desorption/Ionization Time of Flight Mass Spectrometry. (See Revised Manuscript, Page 4. Line 154-155)

Comment 7: Line 197 (Figure 1): Figures a) and b) appear to have been replaced.

Reply to Comment 7: Thank you for your correction. The panels and description in the Figure 1 have been modified. (See Revised Manuscript, Page 5. Figure 1)

Comment 8: Line 216 (Figure 2): Same as previous.

Reply to Comment 8: Thank you for your correction. The panels and description in the Figure 1 have been modified. (See Revised Manuscript, Page 6. Figure 2)

Comment 9: Line 232 (Figure 3): Same as previous.

Reply to Comment 9: Thank you for your correction. The panels and description in the Figure 1 have been modified. (See Revised Manuscript, Page 6. Figure 3)

Comment 10: Line 239 (Table 1): The authors should provide the complete table of ANOVA results with degrees of freedom, sum of squares, mean squares, F and p values.

Reply to Comment 10: Thank you for your kind reminding. We have added the complete table of ANOVA results with degrees of freedom, sum of squares, mean square, F and p values. (See Revised Manuscript, Page 7. Table 1)

Comment 11: Line 445: “6. Patents”???.

Reply to Comment 11: Thank you very much for your kind reminder. We have deleted the section for “Patents”. (See Revised Manuscript, Page 12)

Reviewer 2 Report

The paper “Effects of Heat-processing Conditions on the Digestibility and the Level of Induced Inflammation Levels in the cells of Glycated Glutenin” investigates the effect of processing conditions (temperature, time, pH, mass ratio of reactants) on the digestibility of glycated glutenin. In addition, the effect of reactant mass ratio on the degree of AGE-induced macrophage inflammation was studied.

The study provides interesting insights into the topic, however some aspects require revision.

Line 1: please, specify the type of paper

Abstract

Please, delete the words Background (line 12), Methods (line 15), Results (line 18) and Conclusions (line 22). Author instructions specify these words are not to be specified within the text.

Line 13: please, specify what DCs stands for.

Line 14: please, specify what AGEs stands for.

Line 21: please, specify what MGO stands for.

Introduction

The Introduction is not so linear and I would suggest revising it, so that the reader can get be introduced to the study in a suitable way. Try to fix quickly the context, what has been done so much, what is necessary now, what is the contribution of your study. My suggestion is to revise the style and rephrase sentences with the content you have already put in it.

Lines 29-33: Please, reformulate these sentences so that the meaning is more precise..

Lines 33-34: please, provide reference(s).

Lines 60-62: please, provide the reference number.

Lines 63-64: please, check English language

Materials and Methods

Line 109: “GO” was not defined. In addition, this sentence is not clear, please, reformulate it so that the meaning is clearer.

Line 111: please, replace the comma with a full-stop before the word “Glutenin”

Line 112: please add a comma between 16 and 24.

Line 134: please, replace “Wu, Taylor” with “Wu et al.”

Line 179: “Supplementary Table S1” is not available.

Results

Figure 1: the figures in the two panels do not match the description in the caption. For the a) panel, it is stated that the figure shows glutenin with MGO incubated under different mass ratio of glutenin to MGO, however the figure actually shows GO. The same applies to panel (b). Please, check the figures or the caption.

Line 201: “The molar ratio of protein to AGEs”. The role of this sentence is not clear. Please, amend it.

Figure 2: the figures in the two panels do not match the description in the caption. For the (a) panel, it is stated that the figure shows glutenin with MGO incubated under different mass ratio of glutenin to MGO, however the figure actually shows GO. The same applies to panel (b). Please, check the figures or the caption.

Figure 3: the figures in the two panels do not match the description in the caption. For the (a) panel, it is stated that the figure shows glutenin with MGO incubated under different mass ratio of glutenin to MGO, however the figure actually shows GO. The same applies to panel (b). Please, check the figures or the caption.

Lines 248-250: this sentence sounds like a conclusion or discussion. Please, move it in the following sections.

Figure 4: please, mind that the title for panel (b) and (c) does not match the description in the caption. Is panel (b) showing TNF-α or IL-1β? Is panel (c) showing IL-1β or TNF-α?

Discussion

Line 340: can you provide a reference for the content stated in this sentence, please?

Line 361: check the format of the reference and delete the extra full stop at the end of the sentence.

Line 371: the study you are referring to is Peram et al.

Further changes:

Line 445: please, delete the section for “Patents”, unless you have some information to fix in it.

Line 446: Supplementary Material was not available.

References

Some information (e.g., pages, volume, etc.) are missing for Reference No. 8, 9, 11, 13, 14, 15, 19, 23, 24, 25, 26, 30, 31. Please, amend them.

Author Response

Dear Reviewer:

Thank you very much for your letter and comments on our manuscript entitled “Effects of Heat-processing Conditions on the Digestibility and the Level of Induced Inflammation Levels in the cells of Glycated Glutenin” (ID: foods-1220025). We greatly appreciate the constructive comments that are very helpful for our revision of the manuscript. We have made additions and corrections according to your comments. The revised sections in the manuscript are highlighted in red. We have commissioned native English speakers to revise the language of the manuscript to make it more concise and accurate.

We do hope that the revised manuscript adequately addressed your comments. The major corrections and the response to the reviewers’ comments are as follows:

Comment 1: Line 1: please, specify the type of paper

Reply to Comment 1: Thanks! We are very sorry for our negligence, and we have specified the type of paper in revised manuscript. (See Revised Manuscript, Page 1. Line 1)

Comment 2: Please, delete the words Background (line 12), Methods (line 15), Results (line 18) and Conclusions (line 22). Author instructions specify these words are not to be specified within the text.

Reply to Comment 2: Thank you for your valuable suggestion. We have deleted the words Background (line 12), Methods (line 15), Results (line 18) and Conclusions (line 22). (See Revised Manuscript, Page 1. Line 11-25)

Comment 3: Abstract (Line 13): please, specify what DCs stands for.

Reply to Comment 3: Thank you very much for your kind reminder. “α-DCs” is the abbreviation of α-dicarbonyl compounds. we have specified α-DCs in revised manuscript. (See Revised Manuscript, Page 1. Line 12)

Comment 4: Abstract (Line 14): please, specify what AGEs stands for.

Reply to Comment 4: Thank you very much for your kind reminder. “AGEs” is the abbreviation of advanced glycation end products. we have specified AGEs in revised manuscript. (See Revised Manuscript, Page 1. Line 13)

Comment 5: Abstract (Line 21): please, specify what MGO stands for.

Reply to Comment 5: Thank you very much for your kind reminder. “MGO” is the abbreviation of methylglyoxal. we have specified MGO in revised manuscript. (See Revised Manuscript, Page 1. Line 21)

Comment 6: Introduction: The Introduction is not so linear and I would suggest revising it, so that the reader can get be introduced to the study in a suitable way. Try to fix quickly the context, what has been done so much, what is necessary now, what is the contribution of your study. My suggestion is to revise the style and rephrase sentences with the content you have already put in it.

Reply to Comment 6: Thank you for your valuable suggestion. We have revised the introduction according to your suggestion to make it more linear so that readers can get be introduced to the study in a suitable way. At the end of the second and third paragraphs of the introduction, we add that it is now necessary to study the effects of glycation on glutenin digestibility and effects of glycated glutenin heat-processing conditions on its digestibility. At the end of the study, we add the contribution of this study. (See Revised Manuscript, Page 1. Line 56-57, 69-70, 87-89)

Comment 7: Introduction (Lines 29-33): Please, reformulate these sentences so that the meaning is more precise.

Reply to Comment 7: Thank you for your valuable suggestion. We reformulated these sentences as follows “Wheat is the main food in the human diet and an important source of high-quality plant protein. Glutenin, as an important protein in wheat, possesses high nutritional quality and bioavailability. Heat processing is the main method for wheat products. At high temperatures, the nutrient composition in flour will undergo complex changes. Whilst heat processing will produce certain flavor substances, it will also produce harmful substances that have effects on quality”. (See Revised Manuscript, Page 1. Line 29-34)

Comment 8: Introduction (Lines 33-34): please, provide reference(s)

Reply to Comment 8: Thanks! We are very sorry for our negligence and we have supplemented reference and its citation number in revised manuscript. (See Revised Manuscript, Page 1. Line 33-34)

Comment 9: Introduction (Lines 60-62): please, provide the reference number.

Reply to Comment 9: Thanks! We are very sorry for our negligence and we have supplemented reference number in revised manuscript. (See Revised Manuscript, Page 2. Line 60-62)

Comment 10: Introduction (Lines 63-64): please, check English language.

Reply to Comment 10: Thank you for your kind reminding. We have commissioned Isabela Prendi from the University of Edinburgh to revise the language of the manuscript to make it more concise and accurate. (See Revised Manuscript, Page 2. Line 62-64)

Comment 11: Materials and Methods (Line 109): “GO” was not defined. In addition, this sentence is not clear, please, reformulate it so that the meaning is clearer.

Reply to Comment 11: Thanks! We are very sorry for our negligence, and we have supplemented and revised the manuscript. (See Revised Manuscript, Page 3. Line 96, 112)

Comment 12: Materials and Methods (Line 111): please, replace the comma with a full-stop before the word “Glutenin”.

Reply to Comment 12: Thank you for your correction. We have replaced the comma with a full-stop before the word “Glutenin”. (See Revised Manuscript, Page 3. Line 113)

Comment 13: Materials and Methods (Line 112): please add a comma between 16 and 24.

Reply to Comment 13: Thank you for your correction. We have added a comma between 16 and 24. (See Revised Manuscript, Page 3. Line 115)

Comment 14: Materials and Methods (Line 134): please, replace “Wu, Taylor” with “Wu et al.”

Reply to Comment 14: Thank you for your correction. We have replaced “Wu, Taylor” with “Wu et al.” (See Revised Manuscript, Page 3. Line 138)

Comment 15: Materials and Methods (Line 179): “Supplementary Table S1” is not available.

Reply to Comment 15: Thank you very much for your kind reminder. We have revised the manuscript and added supplementary Material.

Comment 16: Results (Figure 1): the figures in the two panels do not match the description in the caption. For the a) panel, it is stated that the figure shows glutenin with MGO incubated under different mass ratio of glutenin to MGO, however the figure actually shows GO. The same applies to panel (b). Please, check the figures or the caption.

Reply to Comment 16: Thank you for your correction. The panels and description in the Figure 1 have been modified. (See Revised Manuscript, Page 5. Figure 1)

Comment 17: Results (Line 201): “The molar ratio of protein to AGEs”. The role of this sentence is not clear. Please, amend it.

Reply to Comment 17: Thank you very much for your kind reminder. We have deleted “The molar ratio of protein to AGEs”. (See Revised Manuscript, Page 5)

Comment 18: Results (Figure 2): the figures in the two panels do not match the description in the caption. For the a) panel, it is stated that the figure shows glutenin with MGO incubated under different mass ratio of glutenin to MGO, however the figure actually shows GO. The same applies to panel (b). Please, check the figures or the caption.

Reply to Comment 18: Thank you for your correction. The panels and description in the Figure 2 have been modified. (See Revised Manuscript, Page 6. Figure 2)

Comment 19: Results (Figure 3): the figures in the two panels do not match the description in the caption. For the (a) panel, it is stated that the figure shows glutenin with MGO incubated under different mass ratio of glutenin to MGO, however the figure actually shows GO. The same applies to panel (b). Please, check the figures or the caption.

Reply to Comment 19: Thank you for your correction. The panels and description in the Figure 3 have been modified. (See Revised Manuscript, Page 6. Figure 3)

Comment 20: Results (Lines 248-250): this sentence sounds like a conclusion or discussion. Please, move it in the following sections.

Reply to Comment 20: Thank you very much for your kind reminder. We've moved it in the conclusion. (See Revised Manuscript, Page 12. Line 445-447)

Comment 21: Results (Figure 4): please, mind that the title for panel (b) and (c) does not match the description in the caption. Is panel (b) showing TNF-α or IL-1β? Is panel (c) showing IL-1β or TNF-α?

Reply to Comment 21: Thank you for your correction. The title for panel (b) and (c) of Figure 4 have been modified. (See Revised Manuscript, Page 8. See Figure 4)

Comment 22: Discussion (Line 340): can you provide a reference for the content stated in this sentence, please?

Reply to Comment 22: Thank you for your valuable suggestion. We have supplemented reference and its citation number. (See Revised Manuscript, Page 10. Line 347)

Comment 23: Discussion (Line 361): check the format of the reference and delete the extra full stop at the end of the sentence.

Reply to Comment 23: Thank you for your correction. We have deleted the extra full stop at the end of the sentence.  (See Revised Manuscript, Page 11. Line 369)

Comment 24: Discussion (Line 371): the study you are referring to is Peram et al.

Reply to Comment 24: Thank you for your correction. We have modified in revised manuscript. (See Revised Manuscript, Page 11. See Line 378)

Comment 25: Further changes: (Line 445): please, delete the section for “Patents”, unless you have some information to fix in it.

Reply to Comment 25: Thank you for your correction. We have deleted the section for “Patents”. (See Revised Manuscript, Page 12)

Comment 26: Further changes: (Line 446): Supplementary Material was not available.

Reply to Comment 26: Thank you for your correction. We have added supplementary Material.   

RAW264.7 cells (106 cells/well) were placed in a 6-well plate and cultured for 24 hours. After aspirating the medium, fresh medium supplemented with glutenin or glycated glutenin digestion products (10%, v/v) was added to the 6-well plate and cultured for 24 hours. After washing the wells with PBS, the total RNA was extracted with the TRIzo1 (Thermo Fisher Sientific, China) method. The microplate reader is used to determine the purity of the extracted RNA to ensure that the absorbance of the sample is between 1.8-2.1. LunaScriptTM SuperMix Kit (New England BioLabs, Massachusetts, USA) was used for cDNA synthesis. Finally, the genes that encode beta-actin and 18S were used as housekeeping genes. The primer sequences of β-actin, IL-6, TNF-α, IL-1β and RAGE were designed and shown in Supplementary Table S1.

Table 1. The primer sequences of β-actin, IL-6, TNF-α, IL-1β and RAGE

Forward

Reverse

β-actin

ACAGCAGTTGGTTGGAGCAA

ACGCGACCATCCTCCTCTTA

IL-6

CTCTGGCGGAGCTATTGAGA

AAGTCTCCTGCGTGGAGAAA

TNF-α

CGTGGAACTGGCAGAAGAGG

CAGGAATGAGAAGAGGCTGAGAC

IL-1β

AAGGGCTGCTTCCAAACCTTTGAC

TGCCTGAAGCTCTTGTTGATGTGC

RAGE

AGTCCGAGTCTACCAGATTC

CATCTAAGTGCCAGCTAAGG

Comment 27: References: Some information (e.g., pages, volume, etc.) are missing for Reference No. 8, 9, 11, 13, 14, 15, 19, 23, 24, 25, 26, 30, 31. Please, amend them.

Reply to Comment 27: Thank you for your correction. We have revised the manuscript and added information of references. (See Revised Manuscript, Page 13-14)

Round 2

Reviewer 2 Report

Dear Authors,

thank you for amending the manuscript according to minor suggestions.